# CAD Translator: An Effective Drive for Text to 3D Parametric Computer-Aided Design Generative Modeling

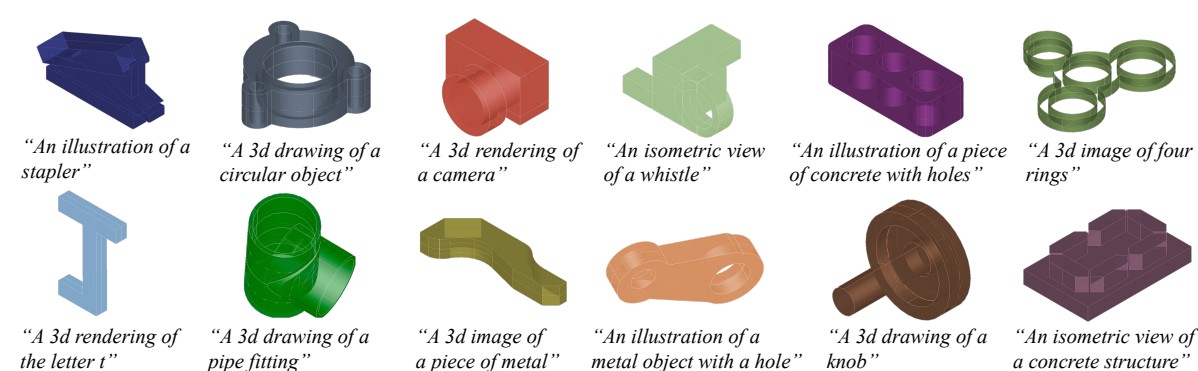

**Figure 1: An overview to show the capability of *CAD Translator*.** Given the text prompt as input, *CAD Translator* would translate them into parametric CAD sequences that can be constructed into 3D shape via designing tools.

## ABSTRACT

Computer-Aided Design (CAD) generative modeling is widely applicable in the fields of industrial engineering. Recently, text-to-3D generation has shown rapid progress in point clouds, mesh, and other non-parametric representations. On the contrary, text to 3D parametric CAD generative modeling is a practical task that has not been explored well, where its shape can be defined with several editable parametric command sequences. To investigate this, we design an encoder-decoder framework, namely *CAD Translator*, for incorporating the awareness of parametric CAD sequences into texts appropriately with only one-stage training. We first align texts and parametric CAD sequences via a *Cascading Contrastive Strategy* in the latent space, and then we propose *CT-Mix* to conduct the random mask operation on their embeddings separately to further get a fusion embedding via the linear interpolation. This can strengthen the connection between texts and parametric CAD sequences effectively. To train *CAD Translator*, we create a *Text2CAD* dataset with the help of Large Multimodal Model (LMM) for this practical task and conduct thorough experiments to demonstrate the effectiveness of our method.

## CCS CONCEPTS

• **Computing methodologies → Computer vision tasks**.

## KEYWORDS

CAD generative modeling, parametric CAD sequence, multi-modal learning

## 1 INTRODUCTION

Computer-Aided Design (CAD) generative modeling plays a crucial role in the fields of design and engineering, providing strong support for manufacturing, visualization, and data management, which drives the advancement of modern design and engineering practices [26, 39, 48, 49]. The CAD model shape design or drawing process can be defined as a parametric CAD sequence of command operations (e.g., line, arc, circle). This kind of representation is called parametric CAD models and can be quickly edited to construct 3D shape. Given its flexibility and practicality, various studies recently have focused on different applications of CAD generative modeling, such as random CAD generation [49], machining segmentation [17], CAD assembly suggestions [16, 47], shape parsing [40], and classification [15].

The parametric CAD model inherently involves two modalities of representation, as it combines textual information (CAD commands and parameters) and (implicit) visual/shape information simultaneously. This means that the execution order of command operations would indicate the process of 3D shape generation. By changing the values of these parameters, the size and shape of the model can be automatically adjusted. Under the current trend of unification of vision and language [35, 41] in the filed of Computer Vision (CV), *text-to-CAD* will be a very interesting problem in the intriguing CAD applications.

In this work, we tackle the problem of leveraging text for parametric CAD generative modeling. An important difference from previous tasks of text-to-image or text-to-3D [1, 19, 25, 30, 38, 42, 50] is that the generated parametric CAD model can be further edited. This is very practical for CAD designers to quickly convert their ideas into coarse-grained CAD models based on text descriptions.

Given parametric CAD models are editable, CAD designers can continue to utilize CAD designing tools to edit and modify these coarse-grained CAD models to obtain the final CAD model, without having to start editing command operations from scratch, accelerating the design process. From another perspective, the parametric sequence is also discrete just like a kind of sentence composed of texts, which builds the foundation for the task of text-to-CAD. As shown in Figure 2, parametric CAD sequences consist of the type of command operations and their corresponding parameters [5, 53]. For each CAD model, parametric sequences can be seen as specific descriptions of its geometry. Executing these parametric sequences sequentially can construct its 3D shape. This is similar to describe features of the object with texts such as "square", "circle", "line" and so on. It motivates us to create the drive to establish a connection between texts and parametric CAD sequences, achieving the generation task of texts to parametric CAD sequences.

Specifically, we design a *CAD Translator* based on an encoder-decoder framework, that effectively incorporates text awareness into parametric CAD sequences. As there is a large gap between the text description and parametric CAD sequences. To this end, we introduce a *Cascading Contrastive Strategy (CCS)* to make them aligned in the latent space. Inspired by mixup [56], we further inject the awareness of parametric CAD sequences into texts via conducting *CT-Mix* to get a new fusion embedding after finishing alignment. Finally, we put all these fusion embeddings into the decoder to recover 3D parametric CAD sequences. By this design, only one-stage training is required to let *CAD Translator* know how to transfer text description into 3D parametric CAD sequences. When it is well trained to learn from texts, *CAD Translator* would have the ability to generate parametric CAD sequences using text as input alone, achieving text to 3D parametric CAD generative modeling easily. Given the text prompt for parametric CAD models is not available in relevant datasets, we first use PythonOCC to render a single image of each 3D CAD model within DeepCAD dataset [49] and then leverage the pretrained CoCa [54] to generate the text description for each parametric CAD model via feeding the rendering image. For more details about the dataset preparation please refer to Sec 5.1.

In summary, our key contributions are as following: (i) We propose a *Cascading Contrastive Strategy (CCS)* controlled by learning steps to align texts and parametric CAD sequences. (ii) We design a *CT-Mix* to incorporate the awareness of parametric CAD sequences into texts and further consolidate *CT-Mix* and *CCS* into a novel multi-modal framework, namely *CAD Translator*, achieving the text to 3D parametric CAD generative modeling. (iii) Extensive experiments demonstrate the effectiveness of our framework on a new dataset with pairs of texts to parametric CAD sequences, namely *Text2CAD*, which is created on one benchmark dataset.

## 2 RELATED WORK

**Parametric CAD Modeling.** We have witnessed significant progress in parametric CAD modeling based on deep learning recently [18, 21, 51]. The graph structure has been used as the representation in CAD for machining feature recognition [4]. BrepNet [20] achieves good performance on the segmentation task of CAD models with operating on B-rep models directly. More recent applications of deep

learning to B-rep models have focused on reconstruction [11, 14]. Besides, several studies starts parsing reverse engineering CAD models. ExtrudeNet [39] designs an effective representation for "sketch-and-extrude" (a common and intuitive modeling process in CAD) to inverse this engineer processing of shape without supervision. SECAD-Net [23] achieves reverse engineering CAD models via learning the implicit sketches and differentiable extrusions from raw 3D shapes as supervision. Furthermore, auto CAD assembling as a practical application has attracted much attention [45, 47]. Unlike these interesting applications based on parametric CAD modeling, our *CAD Translator* is trying to face a challenge of the text to parametric CAD generative modeling, which has not been discussed well. Among of them, ReparamCAD [18] and DeepCAD [49] are most related to our work. DeepCAD essentially focuses on the downstream task of random generation of parametric CAD sequences and ReparamCAD [18] highlights on modifying the style of objects via feeding both the text description and parametric CAD model. Instead, *CAD Translator* is mainly focusing on the generation task of texts to parametric CAD sequences.

**Text-to-3D Shape.** With the success of advanced technology in Large Multimodal Models (LMMs), many inspired applications on text-to-3D shape have raised a surge of interesting from community recently [6, 22, 25, 31, 34]. PointCLIP [58] conducts the alignment between point clouds and 3D category texts via CLIP encoding [35]. CLIP-Mesh [30] present a technique for zero-shot [7, 24] generation of a 3D model with the help of pretrained CLIP. DreamFields [13] optimizes the neural radiance fields (NeRF) [29] for diverse 3D models generation from zero-shot caption with CLIP as guidance. In summary, existing methods either focuses on pretrained LMMs together with distillation to generate 3D shape, or combines with NeRF, or trains a text-conditioned 3D generative model from scratch [6]. The common point of them is that the parametric CAD generation has not been considered yet. Besides, the text-conditioned for parametric CAD modeling has a wide range of applications in the industrial sector. Hence, it motivates us to dive in this interesting task.

**Large Multimodal Models.** Motivated by the desire to boost the unification of language and vision. Large Multimodal Models (LMMs) have drawn significant attention recently. CLIP [35] demonstrates an effective ability to several few-shot tasks via training on a large dataset of image-text pairs. Following works start to combine or revise CLIP-based framework to pursue the better performance on multimodal tasks [3, 37]. The usual approach is to finetune LMMs on a specific task with designing appropriate projection heads or classifiers [2, 36]. Other previous works focus on designing a learnable adapter that can be pluged in LMMs to finetune on only small part of parameters [28, 33, 44]. More recent work 3DALL-E [27] integrates DALL-E [8] into 3D CAD software as a plugin to generate 2D concept maps of 3D objects in the design process. In our *CAD Translator*, we employ LMMs CoCa [54] to generate the description of CAD models for preparing dataset, and make it as a embedding tool for text descriptions.

## 3 PRELIMINARY

For easy to understand the rational design of *CAD Translator*, we first make a brief concept of the parametric CAD sequence. It is a special kind of text with command type and the corresponding

**Figure 2: An overview of the *CAD Translator* method. *CAD Translator* is an autoencoder-based architecture. For the training stage (as shown in the black arrow), the awareness of parametric CAD sequences would be injected into texts via *CT-Mix*. For the inference stage (as shown in the red arrow), only the text description needs to be input to generate the parametric CAD sequence. Finally, these generated parametric CAD sequences can be imported into CAD tools (e.g., PythonOCC and AutoCAD) for visualization or further recreating.**

parameters $(c_t, c_p)$, as shown in Figure 2, where it is always used in the computer-aided design. It allows a designer to modify these parameters to create the object by combining simpler, primitive shapes such as cubes, spheres, and cylinders. The parametric CAD sequence lends itself well to parametric design, where changes in parameters automatically update the model. In this paper, we focus on the generation of single object with command operations including "Line", "Circle", "Arc", and "Extrude". For more details about the definition of the parametric CAD sequence please refer to [49].

## 4  CAD TRANSLATOR

### 4.1  Overview

The *CAD Translator* method is, essentially, a cross-modal autoencoder-based framework with one-stage training. Our method requires a collection of 3D parametric CAD models and associated text descriptions. The overall architecture is shown in Figure 2. During training stage, we first adopt embedding for parametric CAD sequences and texts before feeding them into the encoder. Second, we design a *Cascading Contrastive Strategy* to bring parametric CAD sequences and texts closer after encoding, which is composed of both the constraints of single-modality and cross-modality. Third, we propose *CT-Mix* to achieve the fusion embedding to further inject the awareness of parametric CAD sequences into texts. Finally, the fusion embedding would be fed into the decoder to generate the parametric CAD sequence. Once the network is well trained, the text description as the sole input can go forward to generate the parametric CAD sequence in the inference stage. Following the rule of generated parametric CAD sequences, the 3D shape can be easily achieved and modified by related CAD tools (e.g., PythonOCC and AutoCAD). Please note that labelling the description for parametric CAD models is very time consuming and there is no ready-made text descriptions in existing public CAD datasets. It motivates us to create the text description for each parametric CAD model with the help of CoCa [54]. For more details about dataset preparation

please refer to Sec 5.1. We will release it in the future.

### 4.2  Architecture

**Embedding.** As the usual settings of the transformer-based model, texts and parametric CAD sequences are first projected to an embedding space. For the parametric CAD sequence $(c_t, c_p)$, we imitate the method in [49] to formulate it to an embedding $E_C$ in three aspects:

$$E_C(i) = e_i^{(c_t)} + e_i^{(c_p)} + e_i^{pos}, \tag{1}$$

where $e_i^{(c_t)}$ accounts for the command type $c_t^i$, calculated through $e_i^{(c_t)} = w_{c_t} \delta_i^c$. Here, $w_{c_t} \in R^{d_E \times k}$ is a learnable matrix. $\delta_i^c \in R^k$ denotes $c_t^i$ within the $k$ command types. $e_i^{(c_p)}$ is the embedding of command parameters $c_p^i$, given by $e_i^{(c_p)} = w_{c_p}^a f\left(w_{c_p}^b \delta_i^p\right)$. $f(*)$ flattens the matrix to a vector. Each command is composed of 16 parameters and can be quantized into an 8-bit integer. $w_{c_p}^a \in R^{d_E \times 16 d_E}$ and $w_{c_p}^b \in R^{d_E \times 256}$ are learnable matrices. The function of positional encoding $e_i^{pos}$ is the same as in Transformer [46], which is used to record the index of the command $c_t$ in the complete parametric CAD sequence, In practice, the dimension of $d_E$ is set to 768. For the text $T$, we conduct the pretrained CoCa [54] to encode it to an embedding $E_T$ with the dimension of 768, making it easy to match with $E_C$.

**Encoder and Decoder.** In the training stage, we train an autoencoder with a CAD encoder, a text encoder, and a fusion decoder. For the CAD encoder and the text encoder, there are four layers of transformer blocks with eight attention heads per block and the feedforward dimension is 512. The output of encoders, latent vector dimension is fixed in 256. Based on the same configuration of the CAD encoder and the text encoder, parametric CAD sequences and texts are matched well in terms of dimensions, facilitating cross-modal alignment (*Cascading Contrastive Strategy*) and subsequent knowledge injection (*CT-Mix*). Note that the weights of these two

encoders are independent. Specifically, the encoder $f_{cad}$ and the encoder $f_{text}$ are used to encode $E_C$ and $E_T$ separately. This can be written as:

$$e_{cad} = f_{cad}(E_C), \qquad (2)$$

$$e_{text} = f_{text}(E_T). \qquad (3)$$

The fusion decoder $f_d$ is identical to the encoder in all hyper-parameter settings. One linear layer is connected to with the last block of the fusion decoder to predict parametric CAD sequences $(c_t^*, c_p^*)$, as defined with $f_d(e_{cad}, e_{text}) = (c_t^*, c_p^*)$. When the network is well trained, the text as sole input can generate associated parametric CAD sequences:

$$f_d(f_{text}(E_T)) = (c_t^*, c_p^*). \qquad (4)$$

## 4.3 Cascading Contrastive Strategy

Contrastive learning is widely used to learn representations via attracting positives and repelling negatives [9, 10, 35]. It must be acknowledged that contrastive learning does facilitate the alignment between different modalities for cross-modal learning. However, *CAD Translator* requires a clever design to fully leverage the potential of contrastive learning. We first denote the cross-modal dataset as $D = \{(c_i, t_i)\}$, where $(c_i, t_i)$ denotes a pair of parametric CAD sequence and text description. Our goal is to bring $c_i$ and $t_i$ closer, reducing the gap between them. To go for this, we consider adding both contrastive constraints of the single-modality and the cross-modality simultaneously. For the single-modal contrastive learning on $c_i$, we let each $c_i$ pass forward the CAD encoder $f_{cad}$ twice with dropout under the different rate to generate a pair of positives $(e_{cad}, e_{cad}')$. For each $e_{cad}$, the rest of embeddings $e_{cad}^*$ within one mini-batch are all negatives. Our single-modal contrastive learning aims to catch the similarity within the augmented variants of parametric CAD sequences, making the learned representation preserve the knowledge of $c_i$ comprehensively. For the cross-modal contrastive learning on $(c_i, t_i)$, each $c_i$ is fed into the text encoder $f_{text}$ to obtain $e_{text}$ and paired with associated $e_{cad}$. Similar to the single-modal contrastive learning, $e_{cad}^*$ are as negatives for each $e_{text}$. The cross-modal contrastive learning aims to utilize the knowledge of parametric CAD sequences for better textual feature learning and strike a well connection between them. Finally, the constraints of the single-modality $\mathcal{L}_{C-CAD}$ and the cross-modality $\mathcal{L}_{C-CT}$ can be defined with InfoNCE [32] as following:

$$\mathcal{L}_{C-CAD} = -\mathbb{E}_X \left[ \log \frac{f_k\left(e_{cad}, e_{cad}'\right)}{\sum_{e_{cad}^* \in X} f_k\left(e_{cad}^*, e_{cad}'\right)} \right], \qquad (5)$$

$$\mathcal{L}_{C-CT} = -\mathbb{E}_X \left[ \log \frac{f_k\left(e_{cad}, e_{text}\right)}{\sum_{e_{cad}^* \in X} f_k\left(e_{cad}^*, e_{text}\right)} \right], \qquad (6)$$

where $f_k$ is defined with $e^{\text{sim}(\mathbf{h}_i', \mathbf{h}_i'')/\tau}$. $\text{sim}(*, *)$ denotes the cosine similarity and $\tau$ is a temperature hyper-parameter with 0.05. $X$ denotes the size of one mini-batch during the training.

Inspired by the multi-stage training strategies adopted frequently in machine learning. We propose a *Cascading Contrastive Strategy* (CCS) to split the participation of $\mathcal{L}_{C-CT}$ and $\mathcal{L}_{C-CAD}$ during training stage. The reason is that here lies the conflict: if $\mathcal{L}_{C-CAD}$ and

$\mathcal{L}_{C-CT}$ are activated simultaneously at the beginning of the training, these two constraints are getting some overlaps, making it difficult for model to strike a well balance between them. This would finally hinder the ability of contrastive learning (Recall in Ablation Study 5.2). Hence, *CCS* activates $\mathcal{L}_{C-CT}$ solely from the scratch and incorporates $\mathcal{L}_{C-CAD}$ later, which is effective to alleviate this conflict and fully leverage the capabilities of both two constraints. We utilize the training step as a participating signal for $\mathcal{L}_{C-CAD}$. It can be defined with:

$$\mathcal{L}_{CCS} = \begin{cases} \mathcal{L}_{C-CT} & E \leq S \\ \mathcal{L}_{C-CT} + \mathcal{L}_{C-CAD} & E > S \end{cases}, \qquad (7)$$

where E denotes the current epoch and S is the hyper-parameter of training steps for the adaptive selection of $\mathcal{L}_{CCS}$. The advantage of *CCS* is to make $e_{cad}$ and $e_{text}$ establish a solid connection in early training steps, and then $\mathcal{L}_{C-CAD}$ starts to optimize the learned representation, making it maintain the knowledge of parametric CAD sequences comprehensively. This is quite important for delivering the knowledge of parametric cad sequences to texts.

## 4.4 CT-Mix

Mixup [57] conducts the linear interpolation between two different samples to obtain new augmented data. The detailed implementation can be defined as following:

$$x^* = \lambda x_1 + (1 - \lambda)x_2, \qquad (8)$$

$$y^* = \lambda y_1 + (1 - \lambda)y_2, \qquad (9)$$

where $\lambda$ is sampled from $Beta(\alpha, \beta)$ distribution. $(x_1, x_2)$ denotes the two samples randomly chosen from datasets, and $(y_1, y_2)$ represents the labels of them. $(x^*, y^*)$ is a new sample by linear interpolation. Given its flexibility and friendly implementation, it sparks numerous Mix-based adaptations [43, 52, 55] for operating data augmentation tailored to specific tasks.

The way Mix-based methods construct new data inspires us to transfer it to mix parametric CAD sequences and texts, namely *CT-Mix*. Compared to previous methods, our *CT-Mix* differs in the following two aspects: (i) *CT-Mix* operates the mixing operation between two different modalities rather than focusing on the single modality. (ii) The goal of *CT-Mix* is to inject the awareness of parametric CAD sequences into texts instead of adopting the data augmentation. The advantage of *CT-Mix* is to achieve a fusion embedding $e_{ct}$ that preserves the knowledge of both texts and parametric CAD sequences, further reducing the gap between them. Practically, we conduct the mixing operation between $e_{cad}$ and $e_{text}$ with random masking in the latent space to achieve the fusion embedding $e_{ct}$. Based on Equation 7, This process can be defined as:

$$e_{ct} = \gamma \odot e_{text} + (1 - \gamma) \odot e_{cad}, \qquad (10)$$

where $\gamma$ is a hyper-parameter to control the mixing ratio of $e_{cad}$ and $e_{text}$. In practice, we conduct $\gamma$ as a random 0-1 vector with a setting threshold $R$ (Recall in Hyper-parameter Discussion 5.2) for controlling the ratio of 0 and 1. $\odot$ represents the point-wise multiplication. This is, essentially, a mask operation to combine $e_{cad}$ and $e_{text}$, which can be considered as using $e_{text}$ to fill the masked parts of $e_{cad}$. Under this setting, the awareness of parametric CAD sequences is easily injected into texts. For the same 3D object, $e_{cad}$ and $e_{text}$ are supposed to be its two different representations.

Hence, the directivity of $e_{ct}$ would also aim to the same 3D object. Based on this assumption and Equation 8, the label of $e_{ct}$ can be obtained as:

$$y_{ct} = y_{cad} = y_{text}, \tag{11}$$

which means the labels of them are consistent.

## 4.5 Loss Function

We simultaneously conduct *Cascading Contrastive Strategy* and MSE constraint $\mathcal{L}_M$ to align $e_{cad}$ and $e_{text}$. Then we further adopt *CT-Mix* to achieve the fusion embedding $e_{ct}$. Finally, $e_{ct}$ is fed into the decoder $f_d$ to generate the parametric CAD sequence $(c_t^*, c_p^*)$. To measure the distance from $(c_t^*, c_p^*)$ to $(c_t, c_p)$, we use a standard Cross-Entropy loss $\mathcal{L}_{CE}$. The whole constraint of our model $\mathcal{L}_{CT}$ is defined as following:

$$\mathcal{L}_M = \frac{1}{N} \sum (e_{cad} - e_{text})^2, \tag{12}$$

$$\mathcal{L}_{CE} = -\sum_{i=1}^{N} (c_t, c_p)^i \log(c_t^*, c_p^*)^i, \tag{13}$$

$$\mathcal{L}_{CT} = \mathcal{L}_M + \mathcal{L}_{CCS} + \mathcal{L}_{CE}. \tag{14}$$

The experiments are all trained on one NVIDIA RTX 3090 GPU with a batch size of 256 under 100 epochs. Initial learning rate is set to 0.001 with warm up [12] and gradient clipping of 1.0 is applied in back-propagation.

## 4.6 Metrics

**Accuracy.** As seen in Figure 2, parametric CAD sequences are defined with the command $c_t$ and its parameter $c_p$. [49] proposes to measure the accuracy of the recovered CAD sequence $(c_t^*, c_p^*)$ by calculating $A_C$ and $A_P$ separately. However, The generated parametric CAD sequence with the high accuracy of $A_C$ or $A_P$ may still be failure to construct 3D shape. To make it more reasonable, we add *Sucessful Ratio* ($S_R$) into Accuracy. It is a measurement of the ability to reconstruct the known CAD model, where the input is the existing CAD sequences of the test set. Finally, *Accuracy* (*Acc*) is defined:

$$A_C = \frac{1}{N} \sum_{i=1}^{N} \triangledown[c_t^i = c_t^{i*}], \tag{15}$$

$$A_P = \frac{1}{T} \sum_{i=1}^{N} \sum_{j=1}^{K} \triangledown[c_p^i = c_p^{i*}] \triangledown [\left|c_p^{i,j} - c_p^{i,j*}\right| < \eta], \tag{16}$$

$$S_R = \frac{T_R - F_R}{T_R}, \tag{17}$$

$$Acc = \frac{1}{2} \left[\frac{(A_C + A_P)}{2} + S_R\right], \tag{18}$$

where $\triangledown[*]$ is a boolean function with scalar 0 or 1. $T$ is the total number of parameters in all correctly predicted commands. We set $\eta = 3$ in practice as the error threshold. $T_R$ denotes the total number of predicted CAD sequences. $F_R$ represents the total number of shapes constructed unsuccessfully by predicted CAD sequences.

**Shape Construction.** When the parametric CAD sequence is constructed into 3D shape, we can convert it into point clouds by randomly sampling $K$ points on its surface. In practice, we set $K = 2000$. To measure the differences between a real shape and the predicted shape, we calculate Median Chamfer Distance (MCD) of

them. Furthermore, we also adopt Minimum Matching Distance (MMD) to measure the fidelity of generated shapes with calculating the Chamfer Distance from the 3D shapes in the test set to their nearest neighbors in the generated set. Finally, Jensen-Shanon Divergence (JSD) can be calculated with these converted point clouds in the test set and the generated set, measuring the difference of their data distributions.

**CT-Score.** CT-Score is an important metric to evaluate the similarity of $e_{cad}$ and $e_{text}$. It shows the effectiveness of *CCS* and *CT-Mix* in reducing the gap between parametric CAD sequences and texts, which can guide to determine the appropriate hyper-parameter settings of *CCS* and *CT-Mix* (Recall in Hyper-parameter Discussion 5.2). To achieve this, we calculate the cosine similarity of $e_{cad}$ and $e_{text}$ in the latent space.

$$\cos(\theta) = \frac{1}{N} \sum_{i=1}^{N} \frac{e_{cad}^i \cdot e_{text}^i}{\|e_{cad}^i\| \| e_{text}^i\|}. \tag{19}$$

## 5 EXPERIMENTS

### 5.1 Dataset Preparation

As labelling description for parametric CAD models is very time consuming and the text description is unavailable in existing datasets of CAD parametric models, we move to leverage a pretrained CoCa [54] to generate text for each parametric CAD model. Precisely, we first choose a benchmark dataset, called DeepCAD, which is consist of 178,238 CAD models with their parametric construction sequences [49]. With PythonOCC (OpenCASCADE technology in the Python version), these parametric CAD models can be easily visualized to capture perspective images of them. Specifically, we set position (150, -150, 150) and rotation radian (0.7854, 0.6155, 0.5236) with the setting of front x-axis, right y-axis, and up z-axis as a main viewport for rendering each 3D CAD model. Under this viewport, it ensures capturing as much semantic information of the object as possible in a single image. To complement the visual information missing under the main viewport, we further make the viewport to rotate around the z-axis in $\pi/6$ intervals, from $-\pi/2$ to $\pi/2$ rotation radian. For now, each CAD model is paired with total 7 rendering images with different viewports. Next, we put these perspective images in a pretrained CoCa to generate the text descriptions. Finally, we select the part with the greatest overlap from 7 text descriptions as the final text description that would be paired with each CAD model in DeepCAD dataset. In this way, we have successfully create a new dataset of text to parametric CAD models with the help of a pretrained CoCa and the DeepCAD dataset, namely *Text2CAD*. We adopt the division of *Text2CAD* to obtain 161,240 training pairs, 8,946 validation pairs, and 8,052 testing pairs.

### 5.2 Experimental Results

**Text to Parametric CAD Sequence.** Theoretically, the parametric CAD sequence is akin to the discrete language. These commands and their parameters can be seen as "vocabulary" to form "sentence", which documents the manufacturing process of the 3D object. This formalism gives us the opportunity to leverage language models such as Transformer [46] to achieve this new task. Given the commands in parametric CAD sequences are coupled with different parametric values, setting the parametric CAD sequence apart from

| Method | $A_{CC}\uparrow$ | MCD↓ | MMD↓ | JSD↓ |
|---|---|---|---|---|
| BFSR | 60.13 | 29.65 | 4.56 | 21.64 |
| BFSR (Large) | 62.95 | 26.28 | 3.78 | 17.85 |
| BFSR + LE | 62.71 | 26.79 | 3.84 | 18.07 |
| BFSR + CL | 63.09 | 25.94 | 3.68 | 17.36 |
| BFSR + AUG | 63.72 | 27.76 | 3.92 | 20.09 |
| BFSR (Full) | 64.62 | 24.07 | 3.57 | 16.73 |
| CAD Translator | **70.36** | **21.29** | **2.94** | **10.92** |

**Table 1: The comparison of *BFSR* and *CAD Translator* on the text to parametric CAD sequence generation. $A_{CC}$ is multiplied by 100%. MCD, MMD, and JSD are multiplied by $10^2$. *BFSR* denotes the "brute-force" seq2seq regression strategy that we directly encode texts and decode them into parametric CAD sequences without conducting the CAD related pipeline of *CAD Translator*. (*Large*: increasing the layers of network, *LE*: longer training epochs, *CL*: contrastive learning, *AUG*: data augmentation, *Full*: *Large + LE + CL + AUG*.)**

natural language, as shown in Figure 2. Apparently, there is a noticeable gap in the representation of the same 3D object between parametric CAD sequences and texts. This is telling that, the "brute-force" regression strategy to minimize the difference between them is difficult. To demonstrate this, we first maintain the same configuration of encoder and decoder and remove parametric CAD sequences from the input. Then we make texts go forward to directly approach the parametric CAD sequences, treating it as a "brute-force" seq2seq regression method, namely *BFSR*. Besides, we further adopt some useful learning tricks on *BFSR* to improve its performance (e.g., increasing the layers of network, longer training epochs, contrastive learning, data augmentation.), trying to figure out whether *BFSR* has the potential to catch up with *CAD Translator*. Specifically, we make the following adjustments to BFSR: (i) increasing two layers of encoder and decoder separately, (ii) twice training epochs as *CAD Translator*, (iii) conducting twice dropout in the latent space to generate positive pairs, the same way also used in Equation 6 as a part of *CCS*, (iiii) randomly masking 20% of each embedding in every epoch with a certain probability as the data augmentation. The detail results can be found in Table 1. Note that the shape design is the ultimate goal of CAD models, which means the rationality of parametric CAD sequence is quite important. Our first goal is to generate the valid parametric CAD sequence as much as possible, which can be finally reconstructed into 3D shape. *CAD Translator* outperforms *BFSR* and its variants in all shape construction related metrics and achieves more than about 6% improvement on the accuracy of parametric CAD sequence generation. It demonstrates that the awareness of parametric CAD sequences is injected into texts successfully. Compared to *BFSR*, this further brings texts and parametric sequences closer, making the *CAD Translator* has the strong connection between these two representations. As seen in Figure 3, some key points of final 3D shape generated by *BFSR* is seriously shifted away from Ground Truth (GT) compared to *CAD Translator*. Again, it proves that the task of texts to parametric CAD sequences is challenging, and directly making texts regressed to parametric CAD sequences is difficult.

**Cross Dataset Generalization.** To further validate the generalization capability of *CAD Translator*, we pick up another CAD dataset,

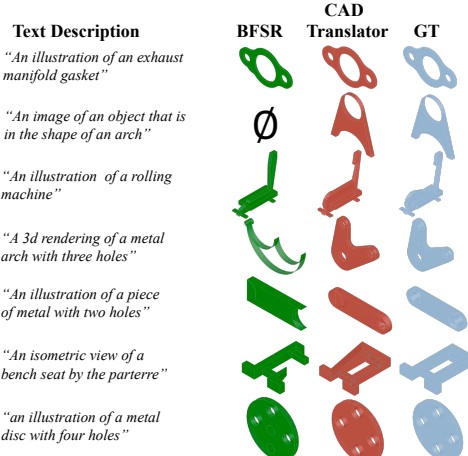

| | BFSR | CAD Translator | GT |
|---|---|---|---|
| **Text Description** | | | |

**Figure 3: Comparison results of texts to parametric CAD sequences on the *Text2CAD* dataset. ∅ denotes the generated parametric sequence that is unable to accomplish shape reconstruction.**

| Method | $A_{CC}\uparrow$ | MCD↓ | MMD↓ | JSD↓ |
|---|---|---|---|---|
| BFSR | 50.38 | 39.52 | 4.63 | 22.20 |
| BFSR (Large) | 51.35 | 37.56 | 3.86 | 19.42 |
| BFSR + LE | 50.85 | 39.36 | 4.03 | 20.49 |
| BFSR + CL | 50.89 | 35.67 | 3.79 | 18.46 |
| BFSR + AUG | 51.41 | 39.43 | 4.45 | 21.25 |
| BFSR (Full) | 51.74 | 35.16 | 3.72 | 17.48 |
| CAD Translator | **56.03** | **32.35** | **3.27** | **12.81** |

**Table 2: Cross Dataset Generalization. Once models are well trained on the *Text2CAD* dataset, they can be tested on the *Text360* dataset.**

Fusion 360 Gallery, which is composed of 2D and 3D parametric CAD models [48]. Then we choose the reconstruction subset of Fusion 360 Gallery and also leverage CoCa [54] to generate text descriptions of parametric CAD models. This is similar to how we construct *Text2CAD* dataset and finally 6,708 samples (*Text360*) are created. Next, we train *BFSR* and *CAD Translator* on the *Text2CAD* dataset and make them tested on the *Text360* dataset directly. The comparable results are as shown in Table 2. It can be found that *CAD Translator* still outperforms *BFSR* and its variants in $A_{CC}$ and all shape construction metrics (especially with a larger margin in $A_{CC}$ and JSD). Compared to *BFSR*, the shape generated by *CAD Translator* is closer to GT and more compatible with the associated text description (Figure 4). It again proves that *CAD Translator* injects the awareness of parametric CAD sequences into texts successfully, making it to learn a robust representation with the capability to adapt in another dataset without additional training. At the same time, it also demonstrates there indeed exists a significant gap between texts and parametric CAD sequences even though they are both some kind of discrete languages. Since the commands and parameters in parametric CAD sequences are fundamentally different entities, it is difficult to make texts approach them directly. Therefore, the "brute-force" method such as *BFSR* cannot effectively

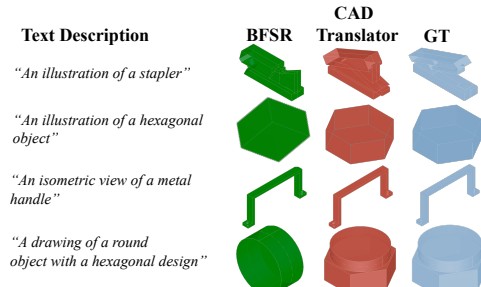

**Figure 4: Comparison results of texts to parametric CAD sequences on the _Text360_ dataset.**

| Method | $A_{CC}\uparrow$ | MCD↓ | MMD↓ | JSD↓ |
|---|---|---|---|---|
| _CAD Translator_ w/o $\mathcal{L}_{\text{C-CT}}$ | 54.18 | 28.72 | 4.25 | 20.03 |
| _CAD Translator_ w/o $\mathcal{L}_{\text{M}}$ | 59.92 | 25.23 | 4.13 | 18.57 |
| _CAD Translator_ w/o $\mathcal{L}_{\text{C-CAD}}$ | 66.91 | 22.45 | 3.49 | 13.68 |
| _CAD Translator_ w/o $\mathcal{L}_{\text{CCS}}$ | 66.84 | 21.87 | 3.28 | 12.51 |
| _CAD Translator_ ∗ | 64.35 | 23.61 | 5.38 | 20.08 |
| _CAD Translator_ | **70.36** | **21.29** | **2.94** | **10.92** |

**Table 3: Ablation study on the _Text2CAD_ dataset. w/o $\mathcal{L}_{\text{CCS}}$ means the parameter $S$ is set to -1 in Equation 7, making $\mathcal{L}_{\text{C-CT}}$ and $\mathcal{L}_{\text{C-CAD}}$ engaged in the entire training process without epoch split. w/o $\mathcal{L}_{\text{C-CT}}$ and w/o $\mathcal{L}_{\text{C-CAD}}$ represent we set $\mathcal{L}_{\text{CCS}} = \mathcal{L}_{\text{C-CAD}}$ and $\mathcal{L}_{\text{CCS}} = \mathcal{L}_{\text{C-CT}}$ respectively.**

address the task of texts to parametric CAD sequences.

| Input (Inference Stage) | Strategy | $A_{CC}\uparrow$ | MCD↓ | MMD↓ | JSD↓ |
|---|---|---|---|---|---|
| _80% CAD + 20% Mask_ | (i) | 89.86 | 3.88 | 1.86 | 4.11 |
| _80% CAD + 20% Text_ | (ii) | 90.75 | 2.65 | 1.79 | 3.88 |
| _70% CAD + 30% Mask_ | (i) | 87.02 | 8.65 | 1.97 | 4.79 |
| _70% CAD + 30% Text_ | (ii) | 88.19 | 4.44 | 1.89 | 4.35 |
| _60% CAD + 40% Mask_ | (i) | 76.25 | 14.02 | 2.07 | 6.75 |
| _60% CAD + 40% Text_ | (ii) | 79.63 | 7.09 | 2.01 | 4.91 |

**Table 4: The results of patching CAD sequences. Note the X% means that X% of tokens in each $e_{cad}$ are retained, and the rest (1-X)% tokens are masked or filled with $e_{text}$.**

**Ablation Study.** We study different settings to figure out the mechanism of _CAD Translator_ and report the results in Table 3. Compared to any weakened version of _CAD Translator_, _CAD Translator_ brings significant improvement to the generation task of texts to parametric CAD sequences. It again proves each component of _CAD Translator_ is indispensable and effective. Especially, when removing $\mathcal{L}_{\text{C-CT}}$, the performance of _CAD Translator_ decays drastically. This also indicates that there is indeed a gap between texts and parametric CAD sequences, and our model effectively reduces this gap. Compared to CAD Translator w/o $\mathcal{L}_{\text{CCS}}$ (means $\mathcal{L}_{C-CT}$ and $\mathcal{L}_{C-CAD}$ are conducted simultaneously), _CAD Translator_ shows a obvious improvement in all metrics, especially over 3% in $A_{CC}$. Besides, we also attempt to let _CCS_ start with only $\mathcal{L}_{\text{C-CAD}}$ and then combine it with $\mathcal{L}_{\text{C-CT}}$, namely _CAD Translator_ ∗. The results

indicate a significant performance degradation when compared to _CAD Translator_ (e.g., about 10% in JSD). It strongly validates that optimizing the learned representation ($\mathcal{L}_{C-CAD}$) prematurely is not a good choice. On the contrary, _CCS_ starts with $\mathcal{L}_{\text{C-CT}}$ and then combine it with $\mathcal{L}_{\text{C-CAD}}$, which can better utilize the contrastive learning of intra-modal and cross-modal to improve the performance of _CAD Translator_.

**Hyper-parameter Discussion on _CT-Mix_ and _CCS_.** To better expose the mechanism of texts to parametric CAD sequences, we conduct several different ratios to combine texts with parametric CAD sequences to get new fusion embeddings with 100 training epochs. Concretely, 20% to 50% as the weight for texts embedding and 80% to 50% as the weight for parametric CAD sequences embedding. Following these weights, different fusion embeddings can be easily generated via conducting _CT-Mix_. Furthermore, we also test $S$ in $\mathcal{L}_{CCS}$ with setting the values of 20 to 50 to explore the potential of _CCS_. Theoretically, _CT-Mix_ and _CCS_ both have the capability to bring texts and parametric CAD sequences as close as possible. Conditioned on this analogy, the more similar $e_{cad}$ and $e_{text}$ are, the more precise the accuracy of texts to parametric CAD sequences generation. Hence, we are trying to find a proper consolidation of _CT-Mix_ and _CCS_ via calculating the _CT-Score_ (Equation 19) of texts and parametric CAD sequences after encoding. Let $R$ denotes the ratio of texts when conducting _CT-Mix_. For example, $R = 20\%$ would result in 20% of 0-1 vector $\gamma$ in Equation 10 is numerical value of 1. $S$ is the hyper-parameter in $\mathcal{L}_{CCS}$ (Equation 7). As shown in Figure 5, it can be inferred that _CAD Translator_ with ($R = 40\%, S = 40$) achieves the highest _CT-Score_ and outperforms other consolidation strategies in all metrics. These comparable results validate this hypothesis, the better _CT-Score_ achieved, the stronger capability to address this task. Besides, we discover that the performance of _CAD Translator_ would decay significantly when $S$ and $R$ are too large or too small, as shown in the four corners of each image within Figure 5. $S$ and $R$ essentially control the degree of participation for texts in the whole training process. Apparently, too much weight of texts makes the _CAD Translator_ take a rough approach like the "brute-force" method in _BFSR_. However, too small weight of texts cannot fully absorb the awareness from parametric CAD sequences, which makes it challenging to bring them closer.

**Synonym Substitution.** To break out of the prompt for each CAD model in the _Text2CAD_ dataset, we conduct synonym replacement on text descriptions to generate similar statements. For example, if the original description of the object starts with "An illustration of...", we substitute its with other descriptions such as "An isometric view of..." or "A description of..." and feed these new prompts into _CAD Translator_ again to show what it can generate. From Figure 6, it can be seen that changing the expression paradigm of describing an object would not create entirely different shapes, especially in only substituting the prefix. More interestingly, when substituting the key word to define the category of a object, it would make some reasonable alterations compared to the original shape (e.g., from "bench" to "long chair" in the second row of Figure 6). This demonstrates that _CAD Translator_ is not limited to the fixed expression and has the ability to generate diverse shapes.

**Patching the parametric CAD sequence.** Patching the parametric CAD sequence is very meaningful for the practical designing conditioned on reusing or recreating the designed entities with

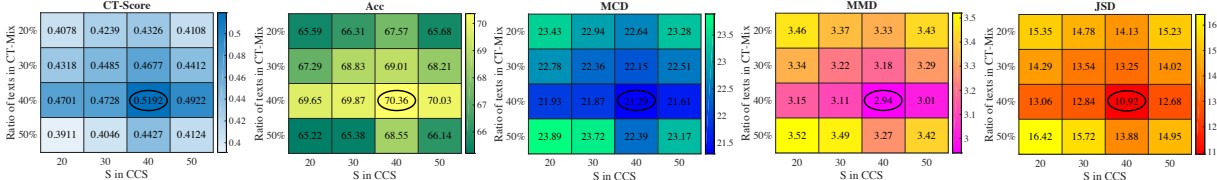

**Figure 5: The comparable experiments on the *Text2CAD* dataset to explore the proper consolidation of *CT-Mix* and *CCS*. *S* represents the training step for the adaptive selection of $\mathcal{L}_{CCS}$ (Equation 7). The y-axis denotes the ratio ($R$) of texts when conducting *CT-Mix*. The black circle in each image highlights the best score achieved by *CAD Translator* in every metric.**

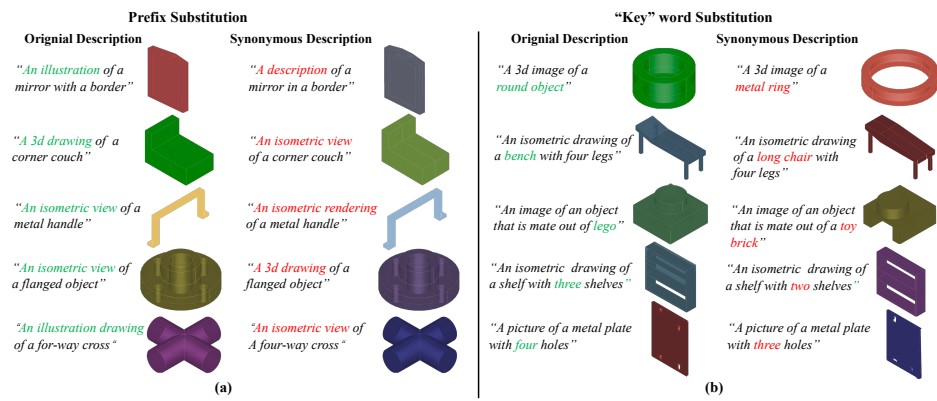

**Figure 6: The synonym substitution results. (a): Prefix Substitution. (b): "Key" word Substitution.**

missing parts of parametric CAD sequences. In addition to achieving the generation task of texts to parametric CAD sequences, *CAD Translator* also can patch the incomplete parametric CAD sequence, where its inputs change to texts and parametric CAD sequences in the inference stage. This is different from the generation task of texts to parametric sequences in which the texts served as the only test inputs. To test the patching capability of *CAD Translator*, we conduct two different strategies for comparing. (i) we let parametric CAD sequences as only inputs and leverage the mask operation with the different rate on them in the latent space. For example, *80% CAD + 20% Mask* means that, for each parametric CAD sequence $e_{cad}$ within test set, 80% of its tokens are retained, and the rest 20% tokens are masked. (ii) compared to (i), we let both parametric CAD sequences and texts as test inputs and set different masks on the $e_{cad}$ where they are filled with $e_{text}$ in the latent space. To be more specific, *80% CAD + 20% Text* means 80% tokens of each parametric CAD sequence $e_{cad}$ are retained and the rest 20% tokens are filled with the associated text $e_{text}$ via our *CT-Mix*. Please note that this application does not require retraining *CAD Translator*. All comparison results are directly tested on *CAD Translator* with hyper-parameters of ($R = 40\%, S = 40$), as shown in Table 4. For strategy (i), we set three levels (60% to 80%) to imitate the missing parts of parametric CAD sequences. Even with only 60% tokens retention of each $e_{cad}$, *CAD Translator* still achieves over 75% $A_{CC}$ of recovering CAD sequences. It proves the effectiveness of our model on patching the incomplete parametric CAD sequence. Besides, compared to strategy (i), strategy (ii) further improves the accuracy of its recovery. Specifically, we also discover that 60% CAD + 40%

Text outperforms 60% CAD + 40% Mask in MCD with the almost 7% improvement. It indicates that *CAD Translator* has a strong ability to patch incomplete parametric CAD sequences with the help of texts. Meanwhile, this also proves that our model dose stick a solid bridge to connect texts and parametric CAD sequences effectively.

## 6 LIMITATIONS

Although *CAD Translator* shows the potential in the generation task of texts to parametric CAD sequences and can provide preliminary CAD modeling for designers, it is still unable to handle more complex CAD models in practical engineering applications. This is because complex CAD models often consist of multiple primitives, resulting in a longer parametric CAD sequence that would exceed the limit command length of 60 in *CAD Translator*.

## 7 CONCLUSION

In this paper, we present *CAD Translator* for automatic text to parametric CAD generative modeling based on transformer network. *CAD Translator* effectively incorporates text awareness into parametric CAD sequences via conducting mixup operation in the latent space, making it possible to generate parametric CAD models with text description under one-stage training. The experimental results verify the effectiveness of our frameworks. Our approach opens up possibilities for leveraging text to parametric CAD generative modeling in the future. To be specific, we will aim to further explore text to parametric CAD modeling in two points: (i) novel ways to combine LMMs, (ii) parametric CAD sequences with text awareness in the latent space.

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
