# OpenReview forum: "CAD Translator: An Effective Drive for Text to 3D Parametric Computer-Aided Design Generative Modeling"
_acmmm.org/ACMMM/2024/Conference — MM2024 Poster_

### Official Review · Reviewer_o3em · 2024-05-16

**Rating:** 4
**Confidence:** 3

**Summary:**

The paper introduces CAD Translator, an approach for text-to-3D parametric CAD generative modeling. It utilizes an encoder-decoder framework to align text descriptions with parametric CAD sequences, employing techniques like Cascading Contrastive Strategy and CT-Mix for enhanced embedding fusion.

**Strengths:**

1. The topic is very interesting, parametric 3D generation is relatively under-explored in the ear of NeRF and Gaussian Splatting.
2. The paper is well-structured and easy to follow.

**Limitations:**

1. Including visual comparisons in the ablation study might better illustrate the effectiveness of L_CSS, which currently appears to be less impactful.
2. The text descriptions generated may not accurately reflect the precise details such as lengths and angles specified in CAD commands, which are crucial for CAD models. It's worth investigating whether the current large language models can comprehend CAD commands as effectively as they do code.
3. The illustrations in the figures could be enhanced for clearer understanding.

**Suitability:**

3

---

### Official Review · Reviewer_PmwM · 2024-05-16

**Rating:** 5
**Confidence:** 3

**Summary:**

The paper propose a framework for text to 3D parametric computer-aided design generative modeling.It first align texts and parametric CAD sequences via a cascading contrastive strategy and then use CT-Mix to  get a fusion embedding. The main contributions are the proposed original framework and its satisfactory effect.

**Strengths:**

1.The paper is well-written, easy to follow, and technically sound.
2.The core idea about  text to 3D parametric computer-aided design generation is original. Few researchers have explored it before. This paper can fill this gap.
3.The performance shows the effectiveness of the proposed method.
4.The qualitative results look impressive.
5.The experiments in the paper are abundant.

**Limitations:**

1.It is better to show some failure cases for simple sentence and some ones with complex description.

The following are some improvements to the paper polish that authors are advised to make in the next edition (especially if accepted)：
1.Ensure each line of text contains a minimum of one-third of its capacity to prevent excessively short lines.
2.Please unify all reference formats.
3.Fig.6 (a)  "for-way"

**Suitability:**

3

---

### Official Review · Reviewer_Nk16 · 2024-05-24

**Rating:** 3
**Confidence:** 3

**Summary:**

This paper investigates a practical task i.e., text to CAD, that aims at generate 3D parametric CAD using textual descriptions. For this task, this paper presents a new framework namely CAD Translator. Specifically, a cascading contrastive strategy (CCS) is introduced, which aligns texts and parametric CAD sequences based on different training steps. In addition, this paper proposes a CT-Mix to incorporate the awareness of parametric CAD sequences into texts. To train and validate the proposed models, this paper repurposes existing datasets and introduce a new benchmark consisting of ~178K text and CAD pairs. Comprehensive experiments validate the effectiveness of the proposed method.

**Strengths:**

+ This paper investigates an interesting and practical task (text2CAD).
+ This paper is well organized and easy to follow.
+ A large-scale dataset with text and CAD pairs is presented based on prior datasets.

**Limitations:**

- The proposed method heavily relies on prior methods [32, 57] with incremental improvements.
- The proposed method seems to be able to only generate some toy CAD results. It would be better to provide some results with more detailed text prompt and showcase some results with fine-grained control capability. The generated CAD models could be useless, if we cannot accurately control or editing the shape of them using textual prompts.
- It would be better to showcase the generated CAD models using videos. It is hard to evaluate the 3D assets using a few images.

**Suitability:**

3

---

### Official Review · Reviewer_DtAJ · 2024-05-24

**Rating:** 5
**Confidence:** 3

**Summary:**

The paper presents a CAD Translator designed on an encoder-decoder framework that can generate parametric CAD sequences solely from text descriptions to assist in the visualization and creation using CAD tools. Through the Cascading Contrastive Strategy (CCS), this framework aligns text descriptions with parametric CAD sequences in the latent space. Then, using the CT-Mix method, the awareness of parametric CAD sequences is infused into the texts to obtain a fused embedding.

**Strengths:**

* Detailed Experiments: The preparation of a detailed dataset, extensive testing, and clear benchmarking thoroughly demonstrate the effectiveness of the CAD Translator. Various metrics have been introduced, confirming that the model performs better than direct methods like BFSR.

* Innovation: The paper addresses the less explored area of text-to-3D parametric CAD generative modeling. By developing the CAD Translator, it achieves direct generation of CAD models from textual descriptions, which is rarely covered in previous research.

*  Clear Structure and Rigorous Logic: The structure of the paper is clear and the logic is rigorous. The methods are explained in detail and are easy to understand, allowing readers to clearly grasp the research results and methodological details.

* Broad Application Prospects: The research results have wide application prospects in the fields of industrial design and CAD software development. The automatic conversion from text to CAD models can significantly enhance design efficiency and lower the barriers to entry in design.

**Limitations:**

* Insufficient Conclusion Content: The current conclusion section is rather brief and does not adequately emphasize the significant findings of the study and its potential application impact. It is recommended that the authors provide a detailed summary of the main achievements of the research in the conclusion, highlight the practical engineering design prospects of this technology (such as the application of repairing parametric CAD sequences), and discuss possible directions for improvement or future research focuses. As mentioned in the limitations, future research could attempt to overcome the length restrictions in the CAD Translator to enable the generation of more complex CAD models.

 * Diversity and Bias: The paper mentions using CoCa to generate descriptions for CAD models, but does not discuss the diversity of the generated descriptions and potential biases. Discussing this issue would help understand how the model performs and its limitations when dealing with diverse data.

* Dataset Dependence: Currently, the training and evaluation of the model primarily rely on the specific Text2CAD dataset, which may limit the model's generalizability in different types of CAD applications.
**  Universality： To verify the universality of the model, future research could consider introducing a diverse range of datasets and testing the model's performance in different data environments.
 **  Generalization Experiments： Its generalizability has only been tested on the Text360 dataset, which is constructed similarly to Text2CAD. Perhaps future experiments with datasets constructed in new and different ways can provide more extensive generalization validation.

* Model Training Computational Costs: The paper does not mention the computational resources and time required for model training. In practical applications, these factors are important indicators for assessing the feasibility of the model. Providing detailed information on these aspects could help evaluate the practicality of the model.

* Lack of Technical Details: Although the paper provides conceptual explanations of the Cascading Contrastive Strategy and CT-Mix, it lacks sufficient implementation details, such as specific algorithm steps or pseudocode. Providing these details would aid in understanding and reproducing the research results.

**Suitability:**

3

---

### Meta-Review · Area_Chair_2vSc · 2024-07-03

**Recommendation:** Accept (Poster)
**Confidence:** 4

**Metareview:**

This paper explores the interesting and practical task of text2CAD, presenting a well-organized and easy-to-follow study. It introduces a large-scale dataset with text and CAD pairs based on prior datasets and delves into the relatively under-explored topic of parametric 3D generation. Overall, the paper is well-structured and effectively addresses an interesting problem within the field.

All reviewers suggest accepting this paper. The AC also recommends acceptance. The authors should revise the paper to address reviewers' comments in the final version.